# Energy Contour Forecasting Optimization with Smart Metering in Distribution Power Networks

**DOI:** 10.3390/s23031490

**Published:** 2023-01-29

**Authors:** Cristian-Dragoș Dumitru, Adrian Gligor, Ilie Vlasa, Attila Simo, Simona Dzitac

**Affiliations:** 1Department of Electrical Engineering and Information Technology, “George Emil Palade” University of Medicine, Pharmacy, Science and Technology of Târgu Mureș, 540088 Târgu Mureș, Romania; 2Distribuție Energie Electrică România Mureș Branch, 540320 Târgu Mureș, Romania; 3Faculty of Electrical and Power Engineering, Politehnica University Timisoara, 300006 Timișoara, Romania; 4Department of Energy Engineering, University of Oradea, 410087 Oradea, Romania

**Keywords:** smart metering, energy measurement, optimization, signal analysis, data series, data analytics, energy contour, real-time processing

## Abstract

Smart metering systems development and implementation in power distribution networks can be seen as an important factor that led to a major technological upgrade and one of the first steps in the transition to smart grids. Besides their main function of power consumption metering, as is demonstrated in this work, the extended implementation of smart metering can be used to support many other important functions in the electricity distribution grid. The present paper proposes a new solution that uses a frequency feature-based method of data time-series provided by the smart metering system to estimate the energy contour at distribution level with the aim of improving the quality of the electricity supply service, of reducing the operational costs and improving the quality of electricity measurement and billing services. The main benefit of this approach is determining future energy demand for optimal energy flow in the utility grid, with the main aims of the best long term energy production and acquisition planning, which lead to lowering energy acquisition costs, optimal capacity planning and real-time adaptation to the unpredicted internal or external electricity distribution branch grid demand changes. Additionally, a contribution to better energy production planning, which is a must for future power networks that benefit from an important renewable energy contribution, is intended. The proposed methodology is validated through a case study based on data supplied by a real power grid from a medium sized populated European region that has both economic usage of electricity—industrial or commercial—and household consumption. The analysis performed in the proposed case study reveals the possibility of accurate energy contour forecasting with an acceptable maximum error. Commonly, an error of 1% was obtained and in the case of the exceptional events considered, a maximum 15% error resulted.

## 1. Introduction

The accurate forecast of electricity consumption is of particular importance in the power system as a whole. From the production of electricity to its consumption, an accurate estimation of the power flow in the national power system leads to minimal losses for both the distribution operators and for the power suppliers, as well as at the lowest possible costs of producing the required power. The price of electricity is also influenced by a precise forecast and if the power supplier has the ability to accurately estimate the amount of power needed in advance, it does not have to purchase electricity at high prices from the next day market [1].

The main actors in the national power system are electricity producers, power suppliers, distribution operators and, last but not least, consumers, who make their own forecasts for the amount of electricity transported within the next day. Wrong planning automatically leads to an excessive or insufficient supply in the energy market, the price of electricity varying from one hour to another depending on the demand and supply of electricity existing in the market at a certain time [2]. Thus, the accurate estimation of electricity from a power system represents an important factor not only for the consumer, but also mainly for the electricity producers, because an overestimation of the electricity needed for the next day primarily affects the green electricity producers, whose production directly depends on the climatic conditions [3,4].

With the increase of electricity demand, the trading process around the world is in accordance with the regulations imposed by the energy market. However, electricity cannot be stored in significant quantities, so there must be a full balance between electricity production and consumption [5]. The demand for electricity depends mainly on economic growth, on the main companies’ turnover, the season, climate factors and unpredictable situations, such as the recent pandemic situation. These unique and specific characteristics lead to a dynamic of the electricity price that can have sudden and unforeseen variations from one day to the next and even from one hour to another, so the electricity supply and distribution companies are forced to intensify the efforts to develop the best possible techniques for forecasting energy requirements [6,7].

Over time, most researchers have focused on point estimates; however, the modernization of the power system and the penetration of smart grids and renewable energy sources has had the effect of increasing uncertainty regarding electricity supply, demand and future prices. Thus, the precise planning of electricity requirements is necessary, with a special emphasis on estimating with high accuracy the power circulated in the entire power system. The actual estimation procedures offer a varied range of possible results depending on the events and uncertainties that may occur during a well established time interval [8,9]. It has been concluded that the amount of electricity needed for the next day as well as its purchase price are more important factors than ever in the planning of daily activities, both for large companies and for home consumers.

Renewable energy sources, such as photovoltaic and wind applications, have grown considerably in recent years, especially by creating prosumers. It is predicted that by 2030, 25% of consumers will become prosumers [10], these efforts being made in order to reduce harmful emissions [11] but also with the aim of reducing energy (e.g., electricity) bills [12]. However, renewable energy sources are in most cases unpredictable sources of energy, their production depending on a lot of uncontrolled factors, mainly meteorological conditions [13,14], thus becoming a problem in accurate estimation models, which contribute to establishing a balanced system from the point of view of the production and sale of electricity.

According to the aforementioned assumptions, the electricity price is directly influenced by the accuracy of the forecast models adopted to estimate the electricity requirements that will be produced and delivered in the national power system in order to cover the demand in a well-established time interval [15], so that there is a balance between the amount of electricity produced and the amount of electricity consumed, everything being related to a reference interval of 15 min.

Based on the aforementioned aspects and on the experience of the authors in terms of involvement and direct confrontation with these issues, this paper proposes and presents a new method developed to solve the important problem of estimating the energy contour for an electricity distribution branch. The main motivation of the proposed approach is to reduce operating costs, while meeting the required performance standards. This is currently necessary not only for compliance with quality standards but especially for the current conditions on the energy market generated by the energy crisis and which require the control of all the variables that compete for safe and sustainable operation. The proposal was developed for electricity utility but it can be extended for other types of utilities where smart metering systems are available.

## 2. Energy Contour in the Context of Smart Metering Availability

In the last decade, power systems have evolved spectacularly, primarily due to the diversified requirements of consumers. The producers and the distribution operators had to adapt to the market requirements, to comply with certain increasingly high standards in terms of the power quality delivered to consumers, standards that were imposed by national and international regulations. These standards define the responsibility of the producers, operators and suppliers of electricity in order to reliably supply industrial or domestic consumers [16].

### 2.1. Smart Metering as a Component of Smart Grid for Energy Forecasting

In recent years, due to the complex demand for data related to the load curve related to each electricity consumer, the implementation of smart systems in the power system was required [17]. Smart grids are the electrical grids that allow a bidirectional power flow [18], which facilitates the possibility of connecting prosumers to the national power system. Being devices that record and communicate the acquired information to a concentrator or directly to a local server [19], smart meters can record both active and reactive–inductive energy, having the possibility to record both electricity consumed from the power system but also that delivered to the system.

In the case of prosumers [20], smart meters can even be programmed with double or even multiple tariffs (e.g., day, night) [21]. Figure 1 presents a typical scheme of a smart metering system suitable for both domestic and industrial consumers. The acquired information is collected in a centralized database and the gathered data can be accessed and processed in real time, thus facilitating the needed electricity estimation process.

Figure 1 highlights the main informational flow mapped on the structure of the electricity contour of a grid related to an electricity distribution company. Substation level metering (1) constitutes the source of information related to the input and exchange with other power networks and the power transmission system. Industrial or commercial metering subsystem (2) along residential/household consumers through the metering subsystem (3) mainly provides the output electricity data from the system. In the future, due to the growth of prosumers and renewable energy sources (RES) adoption, the ratio of input–output electricity flow from the utility grid will evolve significantly. All the data flow that can be used to estimate the electricity contour in real-time is under the supervision of AMI subsystem (4), which communicates with all types of metering subsystems by communication network (5). Communication uses in the field include power line communication (PLC), radio communication (GSM, GPRS, UMTS), DSL, and broadband communication by Ethernet or fiber optic networks.

### 2.2. Energy Contour as Overview Approach vs. Smartgrid

An essential factor in order to deliver electricity with optimal parameters to consumers is to establish the profile of each consumer according to their characteristics. Thus, identification algorithms which are mainly based on centralized data acquired from the smart remote reading systems can be implemented [22]. After establishing the profile of each consumer, a profile is established for each electricity distribution branch, so that purchasing electricity for the next day becomes a simple formality to complete [23]. The consumption profiles can be obtained by inventorying the various household appliances, their power being multiplied by a usage coefficient. Unfortunately, in this case, imprecise data are obtained due to the variety of their uses as well as climatic and cultural factors.

The most appropriate way to establish a consumer’s profile is to collect the data acquired by the smart meters installed at the consumer and electricity transformation points [24]. Thus, a profile can be formed with data measured at every 15 min for different types of consumers such as: industrial, household, economic agents, public lighting, institutions, etc. [25]. By using these profiles and the data acquired from the interconnection points with other distribution operators and with the transport power networks, the profile of any electricity distribution branch [26] can be created. Based on these recorded data, a series of classic, but also modern, algorithms for estimating the energy requirement for the next day can be developed and used [27].

The only impediments to a precise consumer profile determination, based on the data provided by the smart meters, are the small-scale implementations of smart metering systems, the problems related to the confidentiality of the data related to the consumer profile [28], as well as the diversity of existing remote reading systems, each manufacturer of smart meters coming with its own remote reading system [29].

## 3. Electricity Forecasting in Power Distribution Networks

A major challenge in achieving an accurate estimation of electricity is the large volume of data, the existence of the multitude of factors that can influence the production, andthe consumption of electricity [11]. The main factors that contribute to the electricity estimation process are: economic impact, temperature, light and environment factors and social factors, as well as the occurrence of unpredictable events, such as the occurrence of a pandemic situation [30,31].

The electricity forecast can be short-term, which refers to the energy estimate for the following days up to a maximum of two weeks, the amount of energy being reported at hourly or even 15 min intervals [32]. This type of forecast is used in all industrial sectors, as it is necessary to estimate the amount of electricity for any technological and commercial flow. The electricity forecast can also be medium-term, from one month to five years [33,34] and long-term (more than five years) [35]. Each type of forecasting has different particularities and specific implementations, depending on environmental factors, such as meteorological or socioeconomic conditions, which must be taken into consideration [33]. For example, in the first case, temperature, humidity, atmospheric conditions—some considered locally or as average values or as variables correlated with other values—should not be neglected.

The most used method for estimating electricity is represented by linear models, which produce good results if only well-known and very precise parameters are involved in solving the problem, the evolution of the system over time being constant [36]. This method is only suitable for forecasting the electricity needs of a single type of consumer whose profile is well defined. In the event that more complex data are involved and it is necessary to estimate the electricity for a group of consumers, with different fields of activity, it is recommended to use a specially designed algorithm according to the requirements of the problem. This algorithm can be structured with the help of neural networks that offer precise data only if the set of learning data is adequate for the problem. If it is large enough, data from the consumption history of up to five years ago could be necessary [37]. Those kinds of data are quite difficult to obtain for the power distribution operator due to considerations related to the hardware space limit that does not allow keeping the load curves of the consumers for more than 3 years.

The accuracy of electricity estimation is directly influenced by the uncertainty of the data and the interdependence of the variables that enter the forecasting process. These problems can be eliminated by applying a fuzzy model [38], fuzzy logic being successfully used on a large scale in solving problems that have a high degree of uncertainty [39]. Thus, it can also be applied for short-term energy estimation, working much better than the usual forecasting models. However, the estimation of electricity at the level of an electricity distribution branch cannot be correctly forecasted only by the models presented above, because the data involved in the forecasting process vary a lot from one hour to the next and are directly influenced by a multitude of external factors. In this case, it is a necessary use of a hybrid model in which two or even more estimation methods are incorporated.

Due to the intrinsic difficulties of electricity consumption estimation, there are several works dealing with the problem of short-term electricity estimation. For example, in the works [40,41], special emphasis is placed on electricity forecasting with the help of Support Vector Regression (SVR), this being an attractive method due to its adaptability and robustness in dealing with non-linear phenomena. SVM is an automatic learning algorithm that started to be used in the middle of the 20th century [40] in the process of estimating electricity demand. SVM presents some difficulties in tracking non-linear and non-stationary series such as peaks at certain hours of the day or extremes that appear suddenly on a day of the week. In order to obtain good results in these situations as well, Empirical Mode Decomposition (EMD) can be used as an appropriate method especially in non-linear preprocessing and in the analysis of non-linear series. Thus, EMD is effectively used as a preprocessing step for short-term SVR [42].

The paper [43] combines several methods of electricity estimation, such as Empirical Mode Decomposition (EMD), Support Vector Regression (SVR), Particle Swarm Optimization (PSO) and the Econometric Model (ARGARCH), to develop a novel hybrid forecasting model, namely the EMD-SVR-PSO-AR-GARCH model, an algorithm developed to forecast electricity consumption along with economic growth. The presented model takes into account economic and political aspects over a similar reference period, considering both the market economy and the national macrocontrol. The obtained results are quite close to the real electricity consumption curve.

Overall, Table 1 synthesizes recently addressed problems and the approaches used in the field of electricity forecasting. It is shown that, in the last years, researchers have been willing to adopt and develop solutions based on different artificial intelligence techniques and algorithms. Even if the initial approach was given by neural networks, bio-inspired, fuzzy systems and derived algorithms, in the last no more than three years, machine learning has become the preference of many scientific investigators. However, a few developments are still linked to classical approaches, such as regressive based methods, followed by statistical and operational research methods.

Considering the current situation of the energy crisis, not only at the European but also at the global level, together with the previous premises of this situation given by the significant evolution of the role and weight of prosumers, the presence of increasingly unpredictable and significant fluctuations in the energy flow in the power distribution networks can be observed and it is thus necessary to use estimation methods that consider the new conditions that intervene in the process of estimating electricity. The methods developed so far based on time series have demonstrated their efficiency, but in the new context the provided results are limited. An improvement is provided by the methods developed by artificial intelligence techniques, especially in the area of deep-learning. The latter approach for conclusive results depends on the existence of considerable amounts of data and adequate computing power. In many situations, considering the mentioned preconditions, models based on artificial intelligence techniques require retraining periodically or whenever the initial conditions of the use cases evolve significantly. To solve these shortcomings, the present paper proposes a new approach for estimating the electricity contour usable by an electricity distribution company. The authors did not identify such an approach for the mentioned type of application documented in the specialized literature. Apparently, even if the proposed solution is addressed to a particular situation, it can be extended for other use-cases as the proposed model is constituted as a framework in which the constants and parameters of the estimation model are defined as needed. Since a part of these parameters derive from other models, our approach allows the integration of established methods, such as regression models and artificial intelligence algorithms.

This paper is structured in four sections completed by discussions and conclusions. In Section 2, the electricity contour concept of an electricity distribution company that operates under the current regulations found in most European countries and operates modern smart metering systems is defined. Section 3 presents a synthesis of developments in the direction of electricity forecasting in power distribution networks and Section 4 presents the electricity contour forecasting problem formulation related to the scenarios associated with an electricity distribution branch of a national distribution electricity company. After this section, in which the new model and the related proposed algorithm are developed, the numerical results obtained after testing and validating the proposed model are presented through a case study. Finally, the main observations and future development directions are summarized in Section 6.

## 4. Energy Contour Forecasting Problem Formulation

In recent years, due to the increasing diversity and demand for electricity, a special emphasis is placed on the daily realization of the energy contour related to each electricity distribution branch, as well as its estimation for the next day, so that the distribution operator knows in advance the amount of electricity (reported every 15 min) that will enter the system, which will be transferred to neighboring operators and which will be distributed, to both domestic consumers and industrial consumers. Thus, power losses that are directly proportional to the amount of electricity transported in the distribution networks are forecasted.

### 4.1. Framework of Electricity Forecasting

The long-term electricity forecast is one of the most important tasks for an energy supplier but also for a distribution operator in order to purchase electricity at the lowest possible price. In order to estimate the amount of electricity needed to be delivered in the next day to consumers, there are a number of estimation methods. For example, in the work [57], an algorithm that solves the estimation problem with the help of random forests and provides quite good results is presented. The forecasting is directly influenced by the input data, temperature being the main input factor that decisively influences the electricity production forecast, regardless of the day or the season. This method is based on the independent estimation of each hour of the estimation day, each prediction having separate inputs depending on the factors: temperature, the characteristic date and time referred to, as well as the load peaks of the day.

In many situations, external data are coded in historical data such that a proper data-series analysis provides appropriate models for an accurate forecasting. In the particular case of the electricity sector where, due to economic, climate or other factors that influence the trends and seasonal components of electricity contours evolving, the usual forecasting approaches will not be able to follow these changes. To overcome this shortcoming, a framework to solve this challenge is proposed. Figure 2 illustrates the basic principle and stages to be performed, particularized for the case of electricity contour forecasting.

In Figure 2, the chain processing of the AMI data as an input source to the data output (electricity estimated contour) is shown. At the first stage, data are gathered from the AMI system and preprocessed for detecting and filtering the missing or erroneous readings, and are finally consolidated at the distribution branch level. The result consists of a data-series used in the following step to extract, by numerical analysis, frequency domain features. After,  the time-series into the required defined window intervals (hourly, daily, weekly, monthly, seasonal, yearly, etc.) are divided by the extremization of a cost function that ensures the best parameters for estimated patterns. In a continuous flow with new input data, electricity contour estimation can be obtained by using these parameters.

The feature patterns can be updated at each iteration to reduce the computation time and effort on a regular basis established in the updated condition by a specialist.

The feature patterns extraction method is not specified at this stage but any type of classical or machine learning approach can be considered.

### 4.2. Optimization Mathematical Model

The present paper intends to develop and present a hybrid solution for electricity estimation in order to cover the demand for the next day for both industrial consumers and for domestic consumers. The solution consists of the frequency decomposition of the energy contour curve related to a certain analyzed electricity distribution branch for four days, similar to the day for which the estimate is desired. Empirically, it is observed that 11 harmonics result in the case of current requirements on smart metering readings, so the reconstruction of the signal for the estimation day is performed with the help of the 11 amplitudes and phase shifts resulting from an optimization model that includes a series of parameters that directly or indirectly influence the estimation process.

Thus the resulting optimization model is given by the cost function:(1)$$\begin{array}{cc} F(A_{\mathrm{OPf}},A_{\mathrm{OPi}},\varphi_{i}) & =[A_{OPf}-(A_{RD}+k_{1(A_{OPf})}\cdot \left(A_{RD}-A_{1D}\right)+k_{2(A_{OPf})}\cdot \left(A_{RD}-A_{2D}\right)+\\ \; & +k_{3(A_{OPf})}\cdot \left(A_{RD}-A_{3D}\right){)]}^{2}+\lambda_{A}\cdot \sum_{i=1}^{n}\{A_{OPi}-[A_{RDi}+k_{1(A_{OPi})}\cdot \left(A_{RDi}-A_{1Di}\right)+\\ \; & +k_{2(A_{OPi})}\cdot \left(A_{RDi}-A_{2Di}\right)+k_{3(A_{OPi})}\cdot \left(A_{RDi}-A_{3Di}\right){]\}}^{2}+\\ \; & +\lambda_{\varphi }\cdot \sum_{i=1}^{n}\{\varphi_{OPi}-[\varphi_{RDi}+k_{1(\varphi_{i})}\cdot \left(\varphi_{RDi}-\varphi_{1Di}\right)+\\ \; & +k_{2(\varphi_{i})}\cdot \left(\varphi_{RDi}-\varphi_{2Di}\right)+k_{3(\varphi_{i})}\cdot \left(\varphi_{RDi}-\varphi_{3Di}\right){]\}}^{2}, \end{array}$$
subject to: (2)$$\begin{array}{c} A_{OPf},A_{OPi}>0.\\ 0\leq \left|\varphi_{i}\right|\leq 1, \end{array}$$
where:

$A_{OPf}$—fundamental optimal amplitude;

$A_{OPi}$—optimal amplitude of the fundamental harmonics;

$\varphi_{i}$—phase shift between the amplitudes;

$A_{RD}$—amplitude of the reference day (with the first day behind the reference day);

$A_{1D}$—amplitude of the 2nd reference day (with 2 days behind the reference day);

$A_{2D}$—amplitude of the 3rd reference day (with 3 days behind the reference day);

$A_{2D}$—amplitude of the 4th reference day (with 4 days behind the reference day);

$\varphi_{RD}$—reference day phase shift (with the first day behind the reference day);

$\varphi_{1D}$—2nd reference day phase shift (with 2 days behind the reference day);

$\varphi_{2D}$—3rd reference day phase shift (with 3 days behind the reference day);

$\varphi_{2D}$—4th reference day phase shift (with 4 days behind the reference day);

$k_{1}$—correction coefficient related to the first reference day;

$k_{2}$—correction coefficient related to the 2nd reference day;

$k_{3}$—correction coefficient related to the 3rd reference day;

*n*—reference day;

κ—extracted feature (frequency amplitude or phase angle).

The correction coefficients are determined as follows:(3)$$\begin{array}{cc} k_{1(A_{OPi},\varphi_{i})} & =1-\frac{\kappa_{n-1}}{\kappa_{n-2}}\\ k_{2(A_{OPi},\varphi_{i})} & =1-\frac{\kappa_{n-2}}{\kappa_{n-3}}\\ k_{3(A_{OPi},\varphi_{i})} & =1-\frac{\kappa_{n-3}}{\kappa_{n-4}}. \end{array}$$

The optimization problem was modeled starting from the facilities and abundance of data provided by the remote reading systems implemented. The proposed objective is to track in real time the records related to the electricity consumption of each node, comparing them with the records related to the general measurement groups.

### 4.3. Electricity Forecasting Principle

In order to illustrate the proposed electricity forecast methodology, an implementation algorithm was designed. Its main functionalities coded as procedures and an operation mode are illustrated in Algorithm 1.

The key of the proposed algorithm consists of determining the frequency decomposition level features specific for each type of day, with a week, month, season, or a year as a template. By the above-described optimal search algorithm, the main features needed for future specific electricity forecasting on a specific day type are identified. This is accomplished by recomposing the future electricity contour, starting from the template days and their new forecasted feature.

Any signal x(t) can be characterized by two representations:Time domain representation (signal waveform). In this case, the signal x(t) is the representation of the energy contour reported hour by hour for the electricity distribution branch, represented in Figure 3;Representation in the frequency domain (frequency spectrum of the signal). In this case, the signal x(t), which represents the energy contour reported hour by hour for the considered electricity distribution branch, was decomposed into a Fourier series.
**Algorithm 1:** Estimation of energy contour by frequency feature.
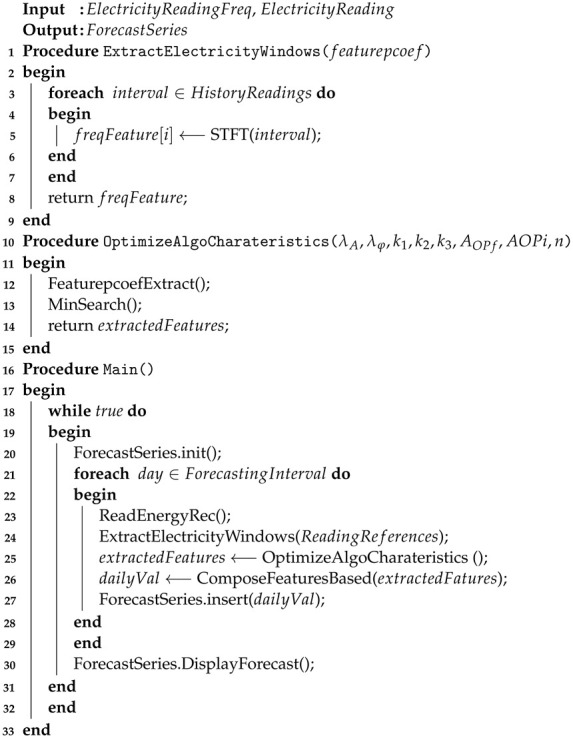


The signal x(t) that represents the energy contour of the analyzed Electricity Distribution Branch, which is periodic for T = 24 h, was developed in Fourier series according to Equation (Equation 4).

The trigonometric form of the signal development x(t) in Fourier series is:(4)$$x\left(t\right)=a_{0}+\sum_{n=1}^{\infty }a_{n}\cdot \cos n\omega_{0}t+b_{n}\cdot \sin n\omega_{0}t,$$
where:

*n*—represents the natural number of harmonics;

$f_{0}=1/T=\omega_{0}/2\pi$—represents the repetition frequency of the signal, also known as the fundamental frequency;

The coefficients $a_{0},a_{n},b_{n}$ can be determined with:(5)$$a_{0}=\frac{1}{T}\int_{T}x\left(t\right)dt;$$
(6)$$a_{n}=\frac{2}{T}\int_{T}x\left(t\right)cos\left(n\omega_{0}t\right)dt;$$
(7)$$b_{n}=\frac{2}{T}\int_{T}x\left(t\right)sin\left(n\omega_{0}t\right)dt.$$

## 5. Energy Contour Forecasting Using Complex Features Components Analysis (ECF) Evaluation—Case Study

In order to test and validate the mathematical algorithm proposed in the paper, the amount of electricity needed for the users of an electricity distribution branch is estimated. The proposed algorithm is tested by using an implementation designed in Matlab R2022b software (The MathWorks, Inc., Natick, MA, USA), both for a working day and a non-working day, during the summer and for the winter season. The results obtained with the ECF method are compared with the results obtained by applying an SVM-type algorithm.

In Figure 4, the electricity forecast for a summer working day is represented using the ECF method.

As can be seen in Figure 4, the estimation error is very small, being different from one interval to another, from a maximum of 10% to just under 1% compared to the real data. The estimation error has both positive and negative values on the entire reference day; an error of less than 1% of the entire amount of electricity that must be purchased for the next day is recorded, around 46 MWh.

Figure 5 presents the electricity forecast for a summer holiday or non-working day by using the ECF method. As can be noticed, the estimation error for a non-working day is more pronounced than for a working day. The error is greater during the period of sudden increase and decrease of the energy contour of the considered electricity distribution branch and it can be observed that, in some intervals, a threshold of over 12% is reached. However, during the studied period, the estimation load curve follows the real one in normal parameters. Overall, the estimation error is below the threshold of 5%.

In Figure 6, the electricity forecast for a winter working day is represented using the ECF method.

As can be seen in Figure 6, even if the load curve of the energy contour is completely modified compared to the one in the summer season, the estimation error is still kept below a mean square deviation of 5% compared to the reference data.

In Figure 7, the electricity forecast for a winter holiday or non-working day is represented using the ECF method.

The load estimation curve in Figure 7 also follows the real load curve, this time without large errors. Even if the energy consumption on the studied day decreased considerably, because a Christmas day was chosen, the estimation method presented in this paper works well even if the electricity fluctuation is quite high from one day to the next. In order to obtain the smallest possible error under these conditions, the training dates were chosen from previous years from days similar to December 25 of the studied year.

In order to validate the results obtained with the estimation algorithm proposed in this work (ECF), which is based on the measurement data acquired in real time from the electricity measurement sensors from each consumer and exchange point, an estimation was made for a period of one year, for each subsequent day separately. It was found that, for each estimated day from the study year, the mean squared error did not exceed 10% of the real data. Electricity estimation was also tested with a method frequently used in the forecasting process, namely the SVM method. In Table 2, the data obtained for the day in which the best results were obtained are summarized, the worst results being obtained when the acquisition sensors provided incomplete or erroneous data, as well as if the electricity fluctuation is significant from one day to the next, as in the situation in Figure 7.

## 6. Conclusions

The reduction of operating expenses resulting from erroneous estimates in the electricity purchase process is an important objective for both the electricity supplier and the distribution operator, who are interested in purchasing a precise amount of electricity for the next day.

Proper decisions especially in near to real-time applications are sustained by specialized custom tools that rely on different types of advanced algorithms. These could suffer from limitations due to known issues, such as low compliance with nonlinearities or very intensive computing resources usage, commonly found in current deep learning solutions. By the proposal introduced in this paper, the main goal was focused on obtaining a solution to overcoming the previously mentioned shortcomings. This is possible in the conditions of maintaining good precision by taking advantage of common features used in common deep-learning algorithms and using them intrinsically to code the model of data without using large artificial intelligence specific data-structures but exploiting the optimal direct search algorithm to provide the most suitable solution.

To validate and highlight the performance of the proposed algorithm, it was tested in order to estimate the energy contour of a power distribution operator for the next day. The proposed algorithm was tested on several types of days, both working days and holidays, in both summer and winter. The obtained results are excellent for working days where the signal format is preserved, the estimation errors being on average ±5% on each reference interval, both positive and negative. The sum of the amounts of electricity erroneously estimated for a day is a maximum of 50 MWh, which represents a maximum of one percent of the amount of electricity circulated in a working day. Less good values are obtained on days when the electricity recording sensors provide incomplete or erroneous values, so that the forecast follows the recording error, or on days when large fluctuations of electricity are registered, such as Christmas Day, where the error on almost every interval is included in the range 8–15%.

Considering the above-mentioned aspects, this paper proposed and designed an original methodology for estimating the electricity contour of an electricity distribution branch. The main advantages of this approach are given by the possibility of providing many inputs of different types to compute the desired results. This implementation is not possible with known classical, or even common, machine learning algorithms. On the other hand, the principles used in machine learning were adopted and integrated in the proposed solution in the form of frequency features that help to characterize the input data in a wider manner.

Deployment of the proposed solution is easily performed for similar problem solving but it is also migrated for applications with a similar background type. This is mainly possible due to the proposed structure, which can be seen as a framework where a particular algorithm that solves sub-problems is possible to enhance with new ones. For example, the feature frequency extraction module in the tested version is based on Fast Fourier Transform. It can also be replaced by a machine learning algorithm which is the next step for future research.

The proposed solution has a few important features. An important one is the power to work with data, for instance when faced with the process of performing research that involves changing the time sampling acquisition due to new rules imposed by regulations of 15 min. intervals vs. 30 min. of historical data.

For many forecasting solutions, especially those that are artificial intelligence based, a reduced amount of historical data leads to their unusability due to improper training or the lack of training completion.

In the case of a large amount of available data, the proposed solution is an economic option due to always storing and operating with the same number of features and consequently with the same, known, amount of needed computation resources. Other solutions, such as machine learning, could, in the same scenario, become an approach a with high demand for computational resources.

Due to the fast and remarkable developments in the field of machine learning, the solution proposed in this paper is intended to be developed using this type of technology.

It will also be investigated as an improvement of optimal features computation by enhancing the optimization model to allow the integration of new parameters or to extends the number of determined parameters for improved final forecasting results.

## Figures and Tables

**Figure 1 sensors-23-01490-f001:**
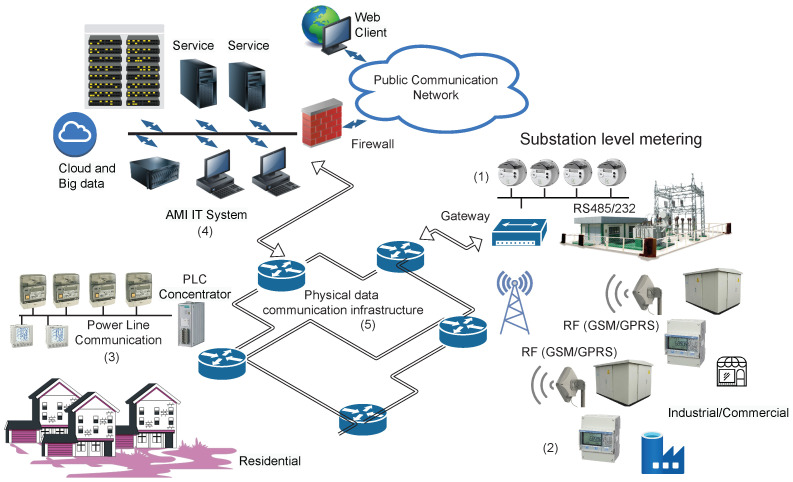
Typical smart metering system infrastructure. Legend: AMI IT System—Analysis Metering Infrastructure IT System; PLC-Concentrator—Power Line Communication Concentrator; RF (GSM/GPRS)—Radio Frequency GSM/GPRS Communication System for Residential/Industrial Smart Meters Group; RS 485/232—Serial Current Loop Communication Network.

**Figure 2 sensors-23-01490-f002:**
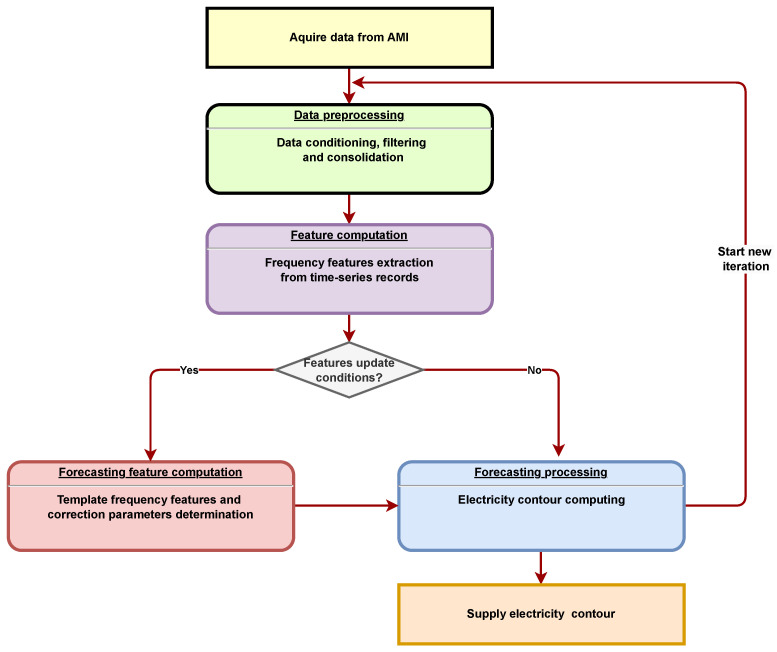
Conceptual framework of the electricity contour forecast.

**Figure 3 sensors-23-01490-f003:**
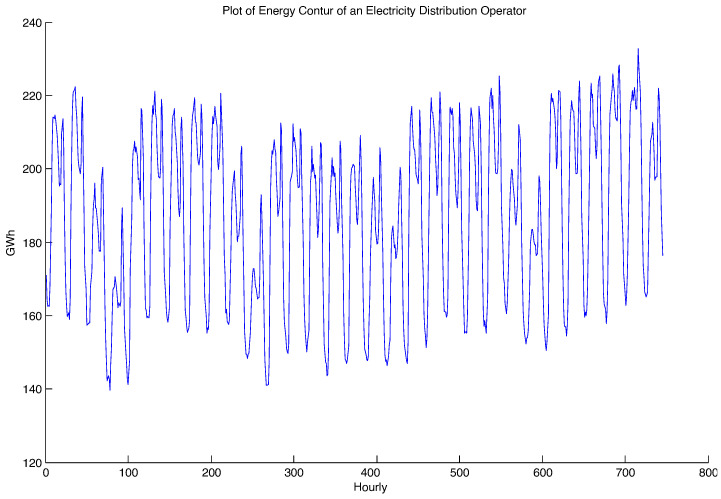
Frequency representation of the energy contour for an Electricity Distribution Branch.

**Figure 4 sensors-23-01490-f004:**
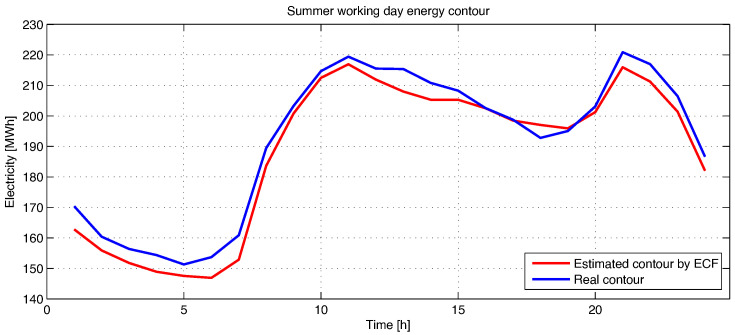
Estimation of energy contour by Frequency Feature Based Method (ECF) for a summer working day.

**Figure 5 sensors-23-01490-f005:**
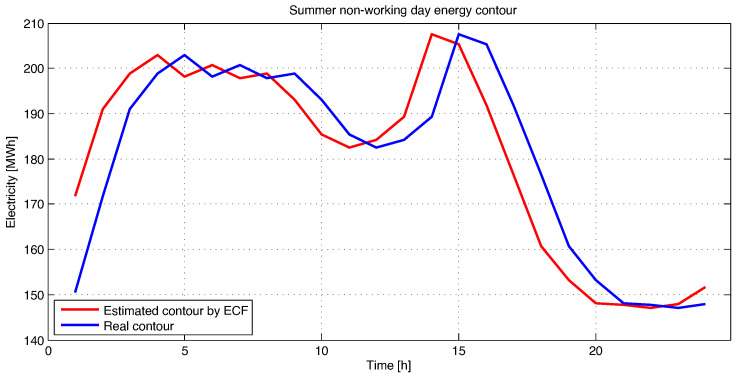
Estimation of energy contour by Frequency Feature Based Method (ECF) for a summer non-working day.

**Figure 6 sensors-23-01490-f006:**
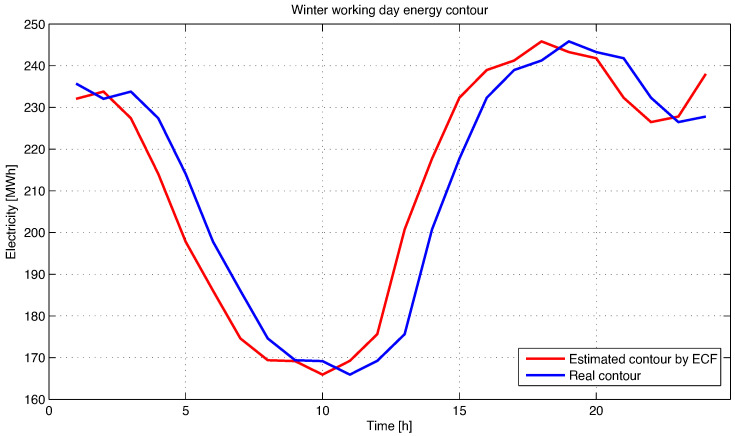
Estimation of energy contour by Frequency Feature Based Method (ECF) for a winter working day.

**Figure 7 sensors-23-01490-f007:**
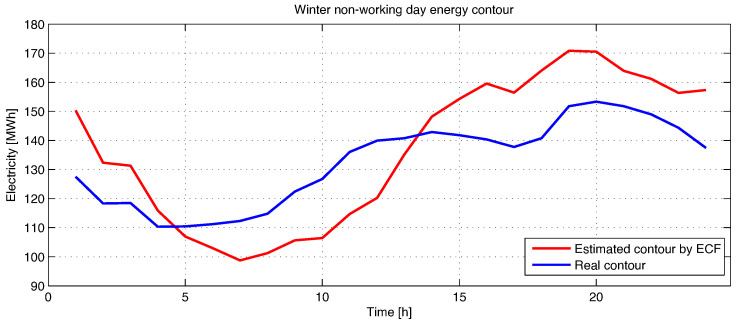
Estimation of energy contour by Frequency Feature Based Method (ECF) for a winter non-working day.

**Table 1 sensors-23-01490-t001:** Electricity forecasting selection and related sources.

Used Approach	Addressed Problem	Relevant References
Regressive, auto-regressive models	Day-ahead electricity price forecasting	[1]
	Electricity demand forecasting	[44,45]
Probabilistic, statistical forecasting or regression analysis	Day-ahead electricity price forecasting	[46]
	Electricity price forecasting	[47,48]
Empirical approach	Power demand forecasting	[49]
Mixed-integer linear programming	Energy provisioning from RES	[50]
Fuzzy logic	Energy provisioning from RES	[39]
Neuro fuzzy model	Long term electricity distribution demand forecasting	[37]
Fuzzy clustering and vague decision-making	Power load forecasting	[38]
Bio-inspired and AI hybrid algorithm	Electricity demand forecasting	[43]
Neural networks and evolutionary computation	Demand side management	[45,51]
	Energy provisioning from non-conventional sources	[52,53]
Machine learning	Day-ahead electricity price forecasting	[54,55,56,57]
	Electricity demand forecasting	[40,41,42,53,58,59,60,61]

**Table 2 sensors-23-01490-t002:** Experimental measured data and optimized results.

Analysis Day	The Amount of Electricity Flow [MWh]	The Amount of Forecasted Electricity by Method 1 [MWh]	The Amount of Forecasted Electricity by Method 2 [MWh]	Method 1 Deviation [%]	Method 2 Deviation [%]
hour 00-01	160,613	134,218	163,069	16.07	−1.52
hour 01-02	153,148	124,566	154,011	12.87	−0.56
hour 02-03	148,114	133,338	147,598	11.43	0.34
hour 03-04	147,070	125,470	145,757	8.44	1.31
hour 04-05	147,025	126,537	145,846	8.36	0.80
hour 05-06	147,860	95,263	150,130	10.42	−1.53
hour 06-07	151,654	113,540	153,984	7.77	−1.53
hour 07-08	172,822	135,765	171,093	5.41	1.01
hour 08-09	191,666	127,062	189,909	6.34	0.91
hour 09-10	198,196	131,865	199,027	9.82	−0.41
hour 10-11	200,227	125,515	201,747	11.01	−0.75
hour 11-12	200,494	118,369	201,787	17.84	−0.64
hour 12-13	201,258	118,369	200,379	17.84	0.43
hour 13-14	200,654	118,369	198,563	17.84	1.04
hour 14-15	194,784	118,369	195,813	17.84	−0.52
hour 15-16	191,297	118,369	194,706	17.84	−1.78
hour 16-17	186,169	118,369	186,001	17.84	0.09
hour 17-18	184,847	118,369	182,207	17.84	1.42
hour 18-19	188,604	118,369	187,107	17.84	0.79
hour 19-20	195,864	118,369	196,303	17.84	−0.22
hour 20-21	209,054	118,369	210,886	17.84	−0.87
hour 21-22	202,487	118,369	202,354	17.84	0.06
hour 22-23	192,127	118,369	189,418	17.84	1.40
hour 23-00	176,431	118,369	175,819	17.84	0.34
TOTAL	4,343,102	1,491,507	156,029	10.46	−0.40

## Data Availability

The data presented in this study are available on request from (I.V.).

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
