# Peer review of "Energy Contour Forecasting Optimization with Smart Metering in Distribution Power Networks"

_sensors, 2023, doi:10.3390/s23031490_

Round 1
Reviewer 1 Report
The author proposes an energy contour forecasting approach. The idea sounds exciting and scientific. The main highlighted points follow such as:
1) The abstract section needs to be more explicit about the envisaged problem, gaps in the literature, and how the contribution solves open issues. A summarization of the main results is missing too. Indeed, combining energy contour forecasting with smart metering can enable more accurate and efficient energy management. For example, by using energy contour forecasts to predict future energy demand, utilities can better match energy production with demand, reducing the need for excess capacity and lowering costs. Furthermore, smart metering can provide real-time data on energy consumption, allowing utilities to fine-tune their energy production and distribution in response to changing demand. Nevertheless, nothing about that is described in the Abstract.
2) Section 2 needs to present the literature gaps and open issues. The reader needs to identify whether the proposal is relevant because the problem needs to be appropriately described. Additionally, as a minor issue, a final paragraph summarizing the remaining sections could be written.
3) Section 2 can be merged with Section 1. Indeed, Figure 1 is a big picture presenting the idea about the envisaged problem.
4) Section 3 is the related works. Given the contemporaneity of the target problem, it would be essential to add a more recent literature discussion. Many very recent solutions are presented in the literature about the same problem. At least 10, 15, or more works (all of them come from 2021, 2022) can be easily found in a systematic review. Contributions based on machine learning algorithms filled the most promising results. A table comparing all works in the literature and the proposal is fundamental to clarifty the innovations claimed by the authors.
5) The core contribution of the paper in Section 4. However, the author needed to be bold regarding the presentation. It is desired to present a big-picture/flow or methodology where the reader can understand all stages used. The reviewer suggests creating a figure with many stages and using further subsections to present each proposal step. It takes much work from the current text version to reproduce the proposal. There needs to detail on how algorithm one works, for example. A typical machine learning algorithm for forecasting could beat the algorithm in terms of performance and accuracy.
6) Section 5 discusses the results. The presentation is chaotic because it was not defined evaluation scenarios or setup configurations. Due to presentation problems in the previous section, it is impossible to understand how the results were obtained. A significant revision in this section is necessary, mainly relating the results to the proposed methodology. Comparison with previous works in the literature needs to be included, too, mainly those supported/based on machine learning approaches.
7) Conclusion is fuzzy because of the problem previously mentioned in the last two sections.
8) Overall, the idea of the paper is interesting but a significant revision is necessary in order to clarify the contributions.
Author Response
Author's response in the attached file.

Reviewer 2 Report
Please check attached file

Author Response
Author's response in the attached file.

Reviewer 3 Report
The paper is interesting and it has potential. However, it needs improvements and clarifications. I can say that the paper needs a better writing and a better structure.
1. The abstract must be rearranged by supporting the numerical values of the obtained results.
2. What is your main contributions and findings in this study? Add a paragraph for relevant discussion in the end of the Introduction.
3. It also lacks statistical tests to verify the significance of the results.
4. What software was used for the solution approach?
5. A comparison with other methods can show the effectiveness of the proposed method.
. What are the advantages of the selected approach? and, what is the disadvantageous of the previously selected methods?
Author Response
Author's response in the attached file.

Round 2
Reviewer 1 Report
I am pleased to report that after a thorough and thoughtful revision process, the authors have attentively considered all suggestions and have made significant improvements to the overall quality of the paper. Therefore, it is my pleasure to recommend approval of the present submission.
Reviewer 2 Report
Accept. All comments well done and
Reviewer 3 Report
The current format is acceptable